# Biology of a putative male aggregation-sex pheromone in *Sirex noctilio* (Hymenoptera: Siricidae)

**Quentin Guignard[1], Marc Bouwer[2¤], Bernard Slippers[3], Jeremy Allison[1,4]**\*

**1** Department of Zoology and Entomology, Forestry and Agricultural Biotechnology Institute (FABI), University of Pretoria, Pretoria, South Africa, **2** Department of Chemistry, Forestry and Agricultural Biotechnology Institute (FABI), University of Pretoria, Pretoria, South Africa, **3** Department of Biochemistry, Genetics and Microbiology, Forestry and Agricultural Biotechnology Institute (FABI), University of Pretoria, Pretoria, South Africa, **4** Natural Resources Canada, Canadian Forest Service, Great Lakes Forestry Centre, Sault Ste. Marie, Ontario, Canada

¤ Current address: Insect Science (Pty) Ltd, Tzaneen, Limpopo Province, South Africa
\* jeremy.allison@canada.ca

**Data Availability Statement:** Data have been uploaded to the Harvard Dataverse (https://doi.org/10.7910/DVN/BHAH1C).

## Abstract

A putative male-produced pheromone has recently been described for the global pest of pines, *Sirex noctilio*, but field-activity has not been demonstrated. This study aimed to investigate the pheromone biology of *S. noctilio* in more detail. Specifically, we i) analysed effluvia and extracts for additional compounds by gas chromatography coupled with electro-antennographic detection (GC-EAD), mass spectrometry (GC-MS) and two dimensional time of flight mass spectrometry (GC X GC TOF MS), ii) conducted dose-response experiments for putative pheromone components, iii) determined the site of synthesis/ storage of the putative pheromone and iv) determined the release rate of the putative pheromone from males and three types of lures. A blend of four compounds was identified, including the previously described (*Z*)-3-decenol and (*Z*)-4-decenol, and two new compounds (*Z*)-3-octenol and (*Z*)-3-dodecenol. All compounds elicited a response from both male and female antennae, but the strength of the response varied according to sex, compound and dose tested. (*Z*)-3-Decenol and (*Z*)-3-octenol at lower and higher doses, respectively, elicited larger responses in males and females than the other two compounds. (*Z*)-3-Octenol and (*Z*)-4-decenol generally elicited larger female than male antennal responses. The site of synthesis and/or storage in males was determined to be the hind legs, likely in the leg-tendon gland. The relative release rate of the major compound by male wasps was shown to be 90 ± 12.4 ng/min, which is between 4 and 15 times greater than that observed from typical lures used previously. These observations are consistent with the hypothesis that these compounds may mediate lek formation in *S. noctilio* males and lek location in females.

## Introduction

The European woodwasp, *Sirex noctilio*, is a global pest of *Pinus* trees. It is native to Eurasia and northern Africa and has been reported worldwide: New-Zealand ~1900 [1], Australia in 1951 [2], South America during the 1980s [3], South Africa in 1994 [4], North America in

**Funding:** This research was funded by the United State Department of Agriculture-Forest Service Forest Health Protection (USDA-FS FHP), National Resources Canada (NRCan), the Tree Protection Cooperative Program (TPCP) and the DSI NRF Center of Excellence in Plant Health Biotechnology (CPHB) in South Africa. The funders had no role in study design, data collection and analysis, decision to publish, or preparation of the manuscript.

**Competing interests:** The authors have declared that no competing interests exist.

2004 [5], and most recently in China in 2013 [6]. Females use a long, stiff ovipositor to drill into the sapwood and inject its eggs, a phytotoxic mucus/venom, and the symbiotic fungus *Amylostereum areolatum* [7]. The mucus/venom facilitates infection by, and growth of, the symbiotic fungus [8, 9] which is important for larval nutrition and survival [10]. Larvae bore into the wood as they develop [11]. The combination of the mucus, fungus and larvae together affect wood quality and often lead to the death of the tree when multiple attacks occur [9, 12].

Control measures for *S. noctilio* focus on silvicultural practices and classical biological control. Several studies have observed that *S. noctilio* preferably attacks stressed and/or supressed trees, and that artificially stressing trees increases attraction of females [13–16]. Removing stressed and supressed trees, thinning and maintaining a low tree density helps to prevent outbreaks of *S. noctilio* [17]. The most widely used biological control agent, *Deladenus siricidicola*, gives variable results in controlling *S. noctilio* populations [18]. Implementation of both silviculture and biological control can lower *S. noctilio* populations and prevent outbreaks, but losses still occur.

Olfactory attractants are important surveillance tools that are used to delineate the distribution, and monitor population dynamics of, *S. noctilio* in pest management programs [19–22]. *Sirex noctilio* is known to detect [23], and be attracted to, a blend of volatiles emitted from stressed host pines [24, 25]. Semiochemical-baited traps are also used to time the application and subsequently measure the efficacy of silvicultural or biological control treatments targeting *S. noctilio* [19, 21, 26]. Volatiles emitted by *A. areolatum* were attractive to the parasitoid *Ibalia leucospoides*, but the compounds involved remain unknown [27]. Although available lures that mimic stressed host pines work reasonably well for high density populations, they do not work well for low density populations of *S. noctilio* [22].

After emerging from the tree, males form leks in the canopy and females must locate these leks for mating although females are parthogenetic and can produce male progeny without mating [15, 28]. Adults do not feed and therefore have a short lifespan of a few days up to two weeks [5, 29], and consequently delayed mating should result in high fitness costs. The existence of a sex pheromone could facilitate rapid mate location and mitigate costs associated with delayed mating. Cooperband et al. [30] identified (*Z*)-3-decenol as the major pheromone released by males, and suggested that (*Z*)-4-decenol and (*E,E*)-2,4-decadienal were additional minor male pheromone components. Laboratory tests demonstrated that the blend of (*Z*)-3-decenol, (*Z*)-4-decenol and (*E,E*)-2,4-decadienal in a ratio of 100:1:1 was most attractive to the wasp. Subsequent field trials observed that pine volatiles were attractive to females and that the addition of the pheromone blend did not increase the number of females captured [25, 31].

The objective of this study was to examine the putative male-produced *S. noctilio* pheromone in more detail. Specifically, we: i) collected male effluvia to confirm the presence of the previously identified putative pheromone components and look for additional components, ii) used electro-antennogram (EAG) analyses to characterise male and female response profiles to two previously identified putative pheromone components [30] and two putative pheromone compounds identified in this study, iii) collected and analysed effluvia of different male body parts to determine the site of production of the putative pheromone and iv) quantified the amount of pheromone emitted by males and compared it to the release rate of three different lures.

## Materials and methods

### Insects

Pine logs infested with *S. noctilio* were collected from Knysna, South Africa during September 2018 and 2019 (n = 133). Trees with characteristic symptoms of infestation, such as browning of needle tips and fresh resin beads on the bark were selected, felled and cut into 80 cm long

logs. Logs were transported to and stored in an insectarium at 20˚C with ambient relative humidity and a photoperiod of 12 hours from late October until the following January. After emergence, insects not used immediately were stored in a refrigerator at 12˚C for later use. Wasps were considered healthy and usable for experiments when they were walking and/or flying within 30 min after being removed from the refrigerator.

## Sample collection

Copper tubing was used to pass dry air through a hydrocarbon trap (Supelco Superpure HC) and the flow rate was controlled with a manual flow rate regulator. The purified air entered a one litre Consol glass jar through Teflon tubing (Supelco, SU20532). Custom made Teflon septa were used as seals for the inlet and an outlet port attached to each Consol jar lid. A glass tube filled with preconditioned PorapakQ (Supelco ORBO 1103, 50/80, 150/75 mg) was connected to the outlet bulkhead union fitting during sampling. Flow rate measurements were made by connecting a glass bubble flow meter to the outlet end of the PorapakQ filled glass traps. Insects were placed in the sealed Consol jars and sampled for 24 hours at room temperature. Jars were washed and oven dried (110˚C) overnight before sampling. After sampling, the PorapakQ inside the glass tubes was placed inside GC vials (vial N9 702293, Macherey-Nigel) and eluted with 1 mL double distilled n-hexane for 2 hours. After 2 hours the solvent was removed and stored in separate 1.5 mL amber storage vials. A total of 42 PorapakQ extracts were obtained from dynamic headspace sampling (30 male, 8 female, 4 male and female).

Insects were individually sampled in a custom made hermetic glass container (0.167 dm$^3$). A Teflon lined septum (Macherney-Nagel 702292) was fitted into an open cap which was screwed onto the top of the hermetic glass container outlet port. A solid-phase microextraction (SPME) fibre (dvb/car/pdms, Supelco 57328-U) was inserted through the Teflon lined septum. The SPME fibre was conditioned at 250˚C for 15 minutes before each use. The glass chamber was partially submerged into a temperature controlled water bath. Insects were sampled individually for 15 minutes at 25˚C. Glass chambers were surface rinsed with 2 mL of double distilled n-hexane after SPME or dynamic headspace sampling. The solvent rinse samples were stored in 1.5 mL amber storage vials and kept in a fridge (4˚C) until analysis. The glass container was washed and dried out in an oven (110˚C) for 30 minutes before each sample collection. A total of 15 SPME samples were obtained from static headspace sampling (11 male, 4 female), and 5 solvent rinse samples were obtained from the glass sampling chambers (4 male, 1 female).

## Sample analysis

**GC-EAD.**   Samples were screened on a GC-EAD system (Agilent 6890N coupled with a Syntech IDAC 4 signal acquisition system) for the presence of electro-physiologically active chromatographic peaks. A total of 76 GC-EAD runs were done with 34 PorapakQ, nine SPME and four glass rinse samples. Samples that showed initial antennal responses were analysed multiple times for confirmation of antennal responses. These included 66 runs done on the ZB-5 column and 10 runs done on the ZB-wax column. A total of 17 different female antennae and 26 different male antennae were used to confirm antennal responses. No antennal preparation was used for more than four GC-EAD runs.

Depending on the type of sample, it was either injected or desorbed splitlessly (vent after one minute at 20 mL/min) through a SPME inlet liner (Supelco 2-6s375,01) at 250˚C. A constant column head pressure of 16 psi (helium) was used. The oven temperature program started at 50˚C for one minute and was increased at a rate of 20˚C per minute up to 300˚C for the ZB-5 column (30 m × 0.320 mm I.D.×0.25 μm film; 7HM-G007-11; Zebron) or 250˚C for the ZB-wax column (30 m × 0.320 mm I.D.×0.25 μm film; 7HM-G002-11; Zebron). The

maximum oven temperature was held for two minutes in each case. A deactivated Y splitter (Agilent 5181–3398) was used to split the column flow at the end of the column to both the EAD and flame ionization detector (FID). The FID detector was heated at 250˚C and the transfer line to the EAD heated to the maximum GC oven temperature used for each column (ZB-wax: 250˚C, ZB-5: 300˚C).

Glass capillaries (Hirschmann 920132) were drawn to a fine tip with a capillary puller (Narishige PP-83). The tips of the capillaries were cut to the required size using a ceramic column cutter. The capillaries were filled with Baedle-Ephrussi-Ringer solution (NaCl: 129, KCl: 4.7, $CaCl_2$: 1.9 millimoles/L). Silver wires were conditioned through electrolysis in HCl (0.1M) using a 9V battery. The conditioned silver wires were inserted in the electrolyte filled capillaries and held in place with the electrode holder. Antennae were removed with a scalpel blade for electroantennographic analyses. The tip of each antenna was cut and connected to the recording electrode while the basal end of the antenna was connected to the reference electrode. Humidified and charcoal-filtered air was used as a carrier gas for the EAD detector. The antennal preparation was placed as close as possible to the EAD outlet. Peaks that were electrophysiologically active were selected and integrated (Chemstation version Rev. B.0211). For those peaks, Kovats retention index (KI) values were calculated from the peak start time. Recorded EAD signals were amplified 10 times and processed with a set of algorithms [32] in RStudio (version 1.1.383).

A calibration curve of (*Z*)-3-decenol was created on the GC-FID system by injecting 1 μL of (*Z*)-3-decenol diluted to 1, 5, 10, 50, 100 and 200 ng/μL in n-hexane. Standards were injected on a GC-FID equipped with a ZB-5 column using the same method previously described. All six diluted standards were injected on the same day. This procedure was repeated over 3 different days.

**GC-MS.** Seven PorapakQ and four SPME samples that were shown to contain male specific chromatographic peaks were analysed by GC-MS (Agilent 7890B coupled to a 5977B MSD). In addition, two glass wash extracts were analysed on the GC-MS. Samples were injected or desorbed splitlessly (vent time at one minute and flow 20 mL/min) through a SPME splitless inlet liner (2–6375.01) at 250˚C. Constant column pressure of 9.8 psi (helium) was used. The oven temperature program started at 50˚C for one minute and was increased at a rate of 20˚C per minute up to 300˚C for the HP-5 MS UI column (30 m × 0.250 mm I.D.× 0.25 μm film; 19091S-433UI; Agilent) or 250˚C for the ZB wax column (30 m × 0.250 mm I. D.×0.25 μm film; 7HG-G007-11; Zebron). Electron impact ionization (70 e⁻V) was used to generate ions. The mass spectrometer ion source was heated to 230˚C and the MS quadrupole was heated to 150˚C. A mass scan range of 45 to 550 m/z at a rate of 4 scan/sec was used for data collection. Peaks that were not in the blank samples were integrated (Chemstation, v. F.01.03.2357) and KI calculated. Tentative identification was done first by comparing the mass spectra of the sample to the NIST library. Then, the KI of the best matched compounds from the NIST library were compared to the KI of the sample. Tentative identities were assigned using the MSD Chemstation (v. F.01.03.2357) software interface and the NIST library (v 2.3).

Dimethyl disulfide (DMDS) was used to derivatize the unsaturated aliphatic chain of the alcohol using a modified version of the protocol described in Buser et al. [33]. The reaction was carried out at 60˚C overnight. Two male samples and one blank sample (no wasp) were derivatized and compared to a derivatized (*Z*)-3-decenol standard. Derivatized samples were injected in the GC-MS with the same method as previously described. Double bond configuration was determined by examining the mass spectrum of the derivatized molecules. Standards were purchased (Advances technology and industrial Co, LTD, Hong Kong) and were serially diluted to 10 ng/μL in n-hexane. Kovats retention index values were calculated and mass spectra of the standards were compared to the selected peaks in our samples to confirm compound

identity. The standard of (*Z*)-4-decenol was injected with (*Z*)-3-decenol at a ratio of 99:1 ((*Z*)-3: (*Z*)-4-decenol) and compared to male extractions injected in the GC-MS.

**GC X GC TOF MS.** Samples that had electrophysiologically active chromatographic peaks were analysed further with GC X GC TOF-MS (Agilent 7890B coupled to a Pegasus LECO). A total of 13 different samples were analysed by GC X GC TOF MS. All the samples that were screened were obtained from PoropakQ extracts. The samples included nine *S. noctilio* male extracts, two *S. noctilio* female extracts and two blank extracts. Samples were injected splitlessly (vent time at 30 seconds and flow at 20 mL/min) through a splitless inlet liner at 250˚C. The primary oven temperature program started at 40˚C for three minutes and was increased at a rate of 10˚C per minute up to 300˚C for five minutes. A Rxi-5 Sil MS column (30 m × 0.250 mm I.D.× 0.25 μm film; 19091S-433UI; Agilent) was installed in the primary oven. A constant flow rate of 1 mL/min (helium) was used. The secondary oven was kept at 5˚C above the primary oven and contained a Rxi-17 Sil MS column (0.97 m × 0.250 mm I.D.× 0.25 μm film; 19091S-433UI; Agilent). The modulator was set at a temperature that was 15˚C more than the secondary oven temperature. The modulation period was set to be 3 seconds with a hot pulse time of 0.8 seconds and a cold pulse of 0.7 seconds. The MS transfer line was heated to 280˚C. Electron impact ionization (-70 e⁻V) was used to generate ions. The mass spectrometer ion source was heated to 230˚C. A mass scan range of 40 to 550 m/z at a rate of 100 spectra/second was used when collecting data. Peaks that could be separated from the major peak on the second dimension and that were not in the blank samples were selected and integrated and compared to the NIST library.

## Electrophysiology

In order to conduct a dose-response experiment we empirically determined the antennal recovery time. Antennae from three males and three females were used and each antenna was used for the same experiment three times. Each experiment started by puffing 10 000 ng of (*Z*)-3-decenol at T = 0. Three puffs of 10 000 ng of (*Z*)-3-decenol were subsequently made on the same antenna after a resting time of T = 30, 45 and 60 seconds from the previous puff. For each experiment, the resting time between two subsequent puffs was randomised. The depolarisation after each resting time was expressed as a percentage compared to the depolarisation at T = 0. No differences were found between male and female recovery time and individuals were pooled together.

Six different concentrations of (*Z*)-3-octenol, (*Z*)-3-decenol, (*Z*)-4-decenol and (*Z*)-3-dodecenol standards were puffed on *S. noctilio* antennae. Each standard was diluted in dichloromethane to 10, 100, 500, 1 000, 5 000 and 10 000 ng/μL. Standards were screened on a total of 10 different male and female antennae. One experiment consisted of a sequence of 10 different puffs with 45 seconds between sequential puffs. Each experiment was bordered by four identical puffs. The first and last puff were solvent blanks (dichloromethane). The second and penultimate puff were control puffs of 10 000 ng of (*Z*)-3-decenol. A trial was considered successful if an antennal response was recorded after puffing each positive control. The six remaining puffs were randomly assigned to the six different doses.

Antennal preparation and electro-antennogram recording were set-up similarly to what was previously described. Antennal preparations were placed in front of an L-shaped glassware linked to a humidified and charcoal-filtered constant air flow (2 m/sec). A 1.5 cm² filter paper (Whatman 1) impregnated with 10 μL of the diluted solution tested was placed in a Pasteur pipette. A Pasteur pipette was linked at one end to a puffed air flow while the other end was inserted inside the glassware directed to the antenna. The air flow directed to the Pasteur pipette was manually controlled via a foot pedal. When the foot pedal was pressed, the sample

tested was puffed for 0.5 sec at 2 m/sec and the constant air flow was cut for 0.5 sec. Depolarisation sizes were directly measured in the Syntech Data Acquisition system for Gas Chromatography with EAD (V4.3).

## Determination of production site of putative male pheromone blend

Volatiles were collected on a SPME fibre as previously described, except that the volatiles were sampled for 15 minutes and the sealed GC vial was placed in a glass chamber in a hot bath at 32˚C. Samples were analysed by GC-FID with the same methodology as described above. Body pieces of one *S. noctilio* were cut using microscissors. Body pieces were left for 20 minutes on a clean laboratory bench to aerate before sampling to allow any cross contamination that may have occurred to evaporate. For the first series of bioassays SPME samples were obtained from the head, thorax, and abdomen separately. Analyses of these samples suggested that the putative pheromone was primarily present in the thorax. The same body segments aerated in the first trial were then further dissected and the wings, front, middle and hind legs, the back half of the thorax (containing the meta- and back half of the meso-thorax) and the front half of the thorax (containing the pro- and front half of the meso-thorax) were aerated in a second series of bioassays. Analyses of these samples mainly detected the putative pheromone components in the hind legs. The same hind legs aerated were further dissected into the femur + coxa, tibia, metatarsus and tarsi and aerated in a third series of bioassays. A total of three male *S. noctilio* were dissected and measured separately.

Standards of (*Z*)-3-octenol, (*Z*)-3-decenol, (*Z*)-4-decenol and (*Z*)-3-dodecenol were injected at a concentration of 10 ng/μL. Peaks that were eluting at the same retention time as the standards were selected and integrated. All body parts from which volatiles were collected were freeze-dried and weighed. The hind legs and thorax could not be directly weighed since they were freshly cut for the prior analyses before being freeze-dried. The weight of the hind legs was calculated as the total weight of the different pieces constituting the hind legs. Similarly, the weight of the thorax was calculated as the total weight of all the cut pieces constituting the thorax. The chromatographic peak areas measured were subsequently divided by the dry mass of the body part measured. The percentage of each compound detected in the different body parts for each series of measurement was calculated. For each series of measurements, results from the three *Sirex* measured were pooled and standardized to an average per individual.

Freeze-dried hind legs of male and female *S. noctilio* were dried overnight in a fume-hood and compared morphologically by scanning electron microscopy (SEM). The tips of the legs were mounted onto aluminium stubs with conductive carbon paint so that the ventral and the dorsal surfaces of the legs were correctly oriented. The sample stubs were then carbon coated using the K950X Turbo Evaporator (Emitech Ltd., UK), before being examined with the SEM (Zeiss Crossbeam 540 FEG SEM, Oberkochen, Germany) at a voltage of 2 kV.

## Comparison of putative pheromone release rates: Males and lures

Preliminary results on the quantification of the putative male pheromone observed very high titres of (*Z*)-3-decenol in effluvia. Release rate data of the rubber septa used in field studies of the putative male pheromone were not reported [25, 31]. We used the same type of red rubber septa as used in previous field trials, and two additional types of lures. The effluvia from three types of lures (silicone ring, polyethylene and red rubber septum) were collected on a SPME fibre and release rates of the (*Z*)-3-decenol were determined by GC-FID. Three lures of each type were filled with 10 μL of (*Z*)-3-decenol (95% purity). The silicone ring (SE) lure consisted of a 3 cm long polydimethylsiloxane rubber tube (2.16 mm OD by 1.02 mm ID) with the ends connected with a 1 cm long capillary glass containing the standard (see [34] for more details).

The polyethylene (PE) lures consisted of the standard loaded in a 0.4 mL capped microcentrifuge tube (Thermo Fisher). Red-rubber septa lures (SEPTA) (Insect science) were soaked in n-hexane for 48 h before use in a sealed jar. The septa were removed from the solvent and the putative pheromone was immediately loaded into each septum and then left in a fume hood for 1 hr to allow the loaded material to be absorbed. One extra lure of each type was not filled with the standard and used as a blank. All lures were put on a cleaned and oven dried (110˚C overnight) metallic grid where each lure occupied a 2 cm$^2$ spot and was separated from the next lure by an empty 2 cm$^2$ square. Lures were left on the plate in a sealed climate controlled room at 23˚C at ambient humidity. All lures were left for two days before the first measurement. Release rates were measured after two and seven days. In addition, three live *S. noctilio* males older than 2 days were measured. Lures and insects were placed in a clean and oven dried hermetic glass jar placed in a 23˚C controlled water bath for measurement. A relative measurement of the release rates of the lures and the insects were performed with a SPME fibre inserted through a septum in the glass jar for 10 min. The SPME fibre was desorbed in the GC-FID inlet and the GC was equipped with a ZB-5 column and the same analysis program was used as previously described.

## Statistical analyses

A Student's t-test test was used to compare the antennal recovery time after confirming that the data were normally distributed (Shapiro test, p = 0.108) and that the variance was homogeneous (Bartlett test, p = 0.228). For the dose-response experiment, solvent blanks were averaged and subtracted from each data point. The control puffs were used to correct for variation in the sensitivity of the antenna, control for any mechanical effects of the airflow and to normalise the data. The difference between the first and second control puff was used to correct for any decrease in antennal sensitivity over time, any observed decrease was assumed to be linear. Corrected antennal responses were expressed as percentages relative to the control puffs. Responses were normalised compared to the control puffs to minimize factors which can affect the absolute size of the response such as the size of the insect [35]. A linear model linking peak area to the quantity of (*Z*)-3-decenol was created. After testing normality of the variable ($\alpha$ = 0.01) and a graphical validation of the residuals, using the "ChemCal" package in R studio [36] the best model was found to be:

$\sqrt{FID\ detector\ response} = 1.12762 \times \sqrt{Z-3-decenol}$, with FID detector response measured in pa × sec and (*Z*)-3-decenol in ng. Data were not normally distributed and a Kruskall-Wallis test ($\alpha$ = 0.05) was used to look for an effect of body part on the amount of the putative pheromone component. Data were compiled and statistical analyses were performed in R Studio V.1.1.383.

## Results

### Sample collection

The chromatograms of samples obtained from female wasps were identical to blank samples. Three chromatographic peaks were found in male samples not present in blank samples. Two compounds were found to elicit antennal responses (Fig 1, Table 1).

### Sample analysis

**GC-EAD.** The first compound to elute was observed in 22 of the 33 male samples screened. Electrophysiological responses from male *S. noctilio* antennae were observed for this chromatographic peak but only rarely and the response was not very strong (a small

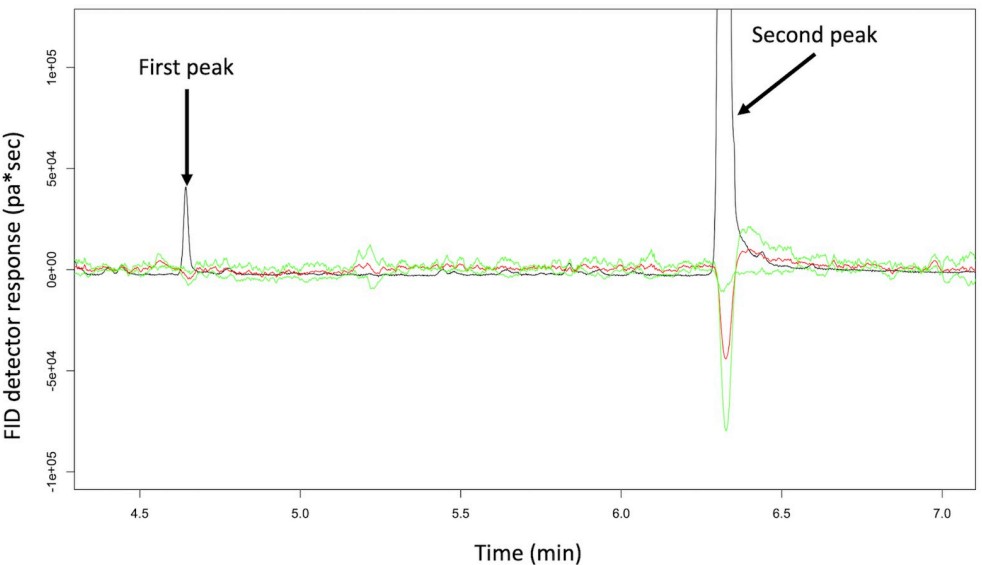

**Fig 1. Processed female antennal responses to the first and second chromatographic peak from male effluvia.**
Black line: FID detector (pa), red line: mean antennal response (µV), green line: standard deviation (n = 6).

depolarisation of approximately 10 µV on 3 out of 12 male antennae and on 1 out of 6 female antenna). The second of the three chromatographic peaks to elute was seen in all 33 samples obtained from males. This peak was the largest peak present in male samples and elicited a response from female (25 µV ± 22 µV, mean ± SD, n = 30) and male (50 µV ± 65 µV, mean ± SD, n = 31) antennae. The third male chromatographic peak was only found in one glass extract. This third peak was not detected in the blank samples or from female glass extract samples. The third chromatographic peak did not elicit an antennal response and was present in much smaller quantities when compared to the other two chromatographic peaks.

 **GC-MS.** After GC-EAD analyses we did not observe any electrophysiologically active peaks in any of the female samples so only male and blank samples were screened on the GC-MS. Of the 14 total runs analysed on the GC-MS, 2 (one glass wash and one male sample) were done on the HP-5 MS UI column and 12 (4 SPME, 7 male extractions and 1 glass wash) were done on the ZB-wax column (Table 1).

 According to the library comparison tentative identities for the first peak could be assigned to three possible isomers of octenol. The ion with the highest m/z in the fragmentation pattern of the first chromatographic peak was m/z = 110 (10). This ion may be created due to the loss of water from octenol. The base peak had a m/z = 55 (100) followed by 81 (58), 68 (50), 67

**Table 1. Kovats retention index values, ratios and antennal response sizes for the three chromatographic peaks found in male extracts (mean ± standard deviation).**

|  | First peak | Second peak | Third peak |
|---|---|---|---|
| KI (ZB-5) GC-EAD | 1055.3 ±1.7 (n = 48) | 1257.1 ± 1 (n = 60) | NA |
| KI (ZB-wax) GC-EAD | 1592.4 ± 0.3 (n = 4) | 1791.6 ± 1.3 (n = 9) | 1997 (n = 1) |
| KI (HP-5 MS UI) GC-MS | 1055 (n = 1) | 1258 ± 0 (n = 2) | 1595 (n = 1) |
| KI (ZB-wax) GC-MS | 1589.4 ± 1.8 (n = 12) | 1789.7 ± 0.8 (n = 12) | 1997.6 ± 1.9 (n = 8) |
| ratio GC-EAD (n = 52) | 1.1 ± 0.6 | 98.9 ± 0.6 | NA |
| ratio GC-MS (n = 9) | 1.1 ± 1.2 | 98.7 ± 1.7 | 0.4 ± 0.6 |
| EAD response | ~10 µV (n = 3 male) | 25 µV ± 22 µV (n = 30 female) | NA |
|  | ~10 µV (n = 1 female) | 50 µV ± 65 µV (n = 31 male) |  |

(44). The first peak was tentatively identified according to the mass spectrum and KI values to three possible compounds: (*Z*)-3-octenol (KI semi polar: 1055 ± 4 (6), KI polar: 1563 ± NA (1)), (*E*)-3-octenol (KI semi polar: 1066 ± NA (1), KI polar: 1550 ± 9 (2)) or (*Z*)-5-octenol (KI semi polar: 1074 ± NA (1), KI polar: 1615 ± 1 (3)).

The second chromatographic peak was tentatively identified as (*Z*)-3-decenol (KI semi polar: 1255 ± NA (1), KI polar: 1789 ± 1 (5)) or (*Z*)-4-decenol (KI semi polar: 1257 ± 1 (7), KI polar: 1791 ± 7 (2)) based on their mass spectra and KI that were most similar on both column types. It was not possible to differentiate between the two isomers based on the mass spectrum.

The identity of the third chromatographic peak was narrowed down to (*Z*)-3-dodecenol and 11-decenol. The highest ion in the fragmentation pattern of the third chromatographic peak was m/z = 166 (5). This ion may be created due to the loss of water from decenol. The base peak had a m/z = 55 (100) followed by 68 (99), 67 (79), 81 (64). The (*Z*)-3-dodecenol (KI semi polar: 1457 ± NA (1), KI polar: 2015 ± 26 (3)) and the 11-decenol (KI semi polar: 1455 ± 12 (2), KI polar: 2023 ± NA (1)) had mass spectra and KI values that were most similar to the second chromatographic peak on both column types.

A small and large chromatographic peak were found in the derivatized male samples. The two derivatized peaks were not detected in the derivatized blank sample. The first small chromatographic peak from the male sample had diagnostic ions m/z = 222 M$^+$ (28), 117 (94) and 105 (23) confirming a double bond at the third carbon of a derivatized octenol molecule. The large second chromatographic peak from the male samples had diagnostic ions m/z = 250 M$^+$ (20), 145 (100) and 105 (30) confirming a double bond at the third carbon of a derivatized decenol molecule. Retention time and diagnostic ions in the derivatized (*Z*)-3-decenol standard were identical to the large second chromatographic peak found in the derivatized male sample. Diagnostic ions included m/z = 250 M$^+$ (21), 145 (100) and 105 (30).

**GC X GC TOF MS.**   All male samples screened on the GC X GC TOF MS contained a large peak not present in blanks that eluted between 450 and 465 seconds in the first separation dimension. The second dimension revealed a small (A) and large (B) chromatographic peak that were not present in blank samples (S1 Fig). The large peak was present in very high quantity compared to the small peak and overloaded the column. Consequently, the small peak elutes in the tail of the large peak. The two peaks start eluting at T = 1.0 and 2.0 seconds on the second dimension for peaks A and B. The best matches for peak A in the NIST library include six isomers of decenol. The best results for the large co-eluting peak B in the NIST library include seven isomers of decenol. For both co-eluting peaks A and B the two best matches in the library are (*Z*)-3-decenol and the (*Z*)-4-decenol. The ratio of these two peaks was calculated to be 98.1 ± 0.6: 1.9 ± 0.6 (A: B, mean ± SD, n = 9).

## Compound identification

Retention index value and mass spectral comparison with a standard confirmed the first chromatographic peak was (*Z*)-3-octenol. Vinogradov [37] reported that (*E*)-3-octenol and (*Z*)-3-octenol are separated by 5 and 23 KI units on a non-polar and polar column respectively. A difference of 5 and 23 KI is feasible in our set-up, and the (*Z*)-3-octenol standard elutes at the same retention time as the peak in our sample. The (*Z*)-3-octenol standard had a KI (semi polar: 1056, polar: 1591) and mass spectrum (55 (100), 67 (44), 68 (47), 81 (58), 82 (19), 95 (13), 110 (10)) that was similar to the first peak found in the male samples (KI semi polar 1055.3 ±1.7, polar 1592.4 ± 0.3 mass spectrum: 55 (100), 67 (44), 68 (50), 81 (57), 82 (19), 95 (13), 110 (10)). The PubCHem library indicates that the three major ions for (*E*)-3-octenol were m/z = 55, 41 and 68 (from highest to lowest), and m/z = 55, 81, 41 for the (*Z*)-3-octenol. The most abundant ion after m/z = 55 in our samples is the m/z = 81 and not the m/z = 68

(ion m/z = 41 was out of range of the detector set-up), suggesting that the peak is (*Z*)-3-octenol rather than the (*E*)- isomer.

The identity of the large second chromatographic peak was confirmed to be (*Z*)-3-decenol. The (*Z*)-3-decenol standard elutes at the same retention time as this peak in our samples. In addition, Tamura et al. [38] reported that (*E*)-3-decenol and (*Z*)-3-decenol are separated by 6 and 25 KI units on an non-polar and polar column, respectively. The (*Z*)-3-decenol standard had a KI (semi polar: 1255, polar: 1788) and mass spectrum (55 (100), 67 (73), 68 (83), 81 (57), 82 (35), 95 (23), 96 (21)) that was similar to the large second chromatographic peak found in male samples (KI semi polar 1257 ±1; polar 1792 ± 1; mass spectrum: 55 (100), 67 (73), 68 (85), 81 (57), 82 (34), 95 (22), 96 (17)).

The identity of the small co-eluting chromatographic peak is likely (*Z*)-4-decenol. Derivatized samples did not show diagnostic ions specific to a double bond in the fourth carbon position, likely due to a very small amount of the compound in the samples analysed. Chromatographic peaks corresponding to the standards of (*Z*)-3 and (*Z*)-4-decenol in a ratio of 100: 1 ng/μL could not be separated on the GC-MS. Samples analysed in the GC-MS gave similar results as the standards tested in a 100:1 ratio, where the (*Z*)-4-decenol chromatographic peak elutes in the tail after the large chromatographic peak of (*Z*)-3-decenol. (*Z*)-3-Decenol was ~100–1000 times more abundant than (*Z*)-4-decenol which overloaded the column. For this reason, separation of (*Z*)-3-decenol and (*Z*)-4-decenol extracted from the male sample was not possible by GC-MS in this study.

The identity of the third chromatographic peak could not be confirmed by derivatisation. The (*Z*)-3-dodecenol standard elutes at the same retention time as the peak in our sample. In addition, Marques et al. [39] reported that (*E*)-3-dodecenol and (*Z*)-3-dodecenol were separated by 6 and 24 KI units on the DB-5 and DB-wax columns, respectively. The standard of (*Z*)-3-dodecenol had a KI (semi polar: 1457, polar: 1996) and mass spectrum (ion abundance: 55 (99), 67 (76), 68 (94), 82 (60), 95 (28), 96 (28)) very similar to the third peak found in male samples (KI semi polar 1595, KI polar 1997.6 ± 1.9 mass spectrum: 55 (99), 67 (79), 68 (97), 81 (64), 82 (58), 95 (28), 96 (26)). The KI and mass spectrum strongly suggest that the third chromatographic peak identity is the (*Z*)-3-dodecenol.

## Electrophysiology

No differences were observed between the size of the depolarisation after 45 and 60 seconds (S2 Fig). Therefore, we used an interval between the two puffs of 45 seconds in the dose-response experiment. (*Z*)-3-Octenol elicited a significantly larger relative response than the other compounds tested at doses > 5 000 ng ($p < 0.05$, Fig 2a). Females were more sensitive than males to the (*Z*)-3-octenol except for doses of 100 ng ($p = 0.358$) and 10 000 ng ($p = 0.104$). Similar antennal responses were recorded for doses between 50 000 ng and 100 000 ng of (*Z*)-3-octenol for both males ($p = 0.344$) and females ($p = 0.520$), indicating antennal saturation.

Among all the putative pheromone components, (*Z*)-3-decenol elicited the biggest relative responses for doses < 5 000 ng and second biggest relative response for doses > 5 000 ng (Fig 2b). For each dose tested, both males and females had similar relative responses (all $p > 0.05$). The antennal responses were similar when doses of 50 000 ng and 100 000 ng were puffed for both males ($p = 0.912$) and females ($p = 0.734$), indicating saturation of the antenna for this compound.

(*Z*)-4-Decenol was found to elicit similar responses as (*Z*)-3-octenol for doses < 10 000 ng (Fig 2c). Females were more sensitive to (*Z*)-4-decenol than males except when 100 ng ($p = 0.558$) or 100 000 ng ($p = 0.104$) were puffed. As with the two previous compounds tested,

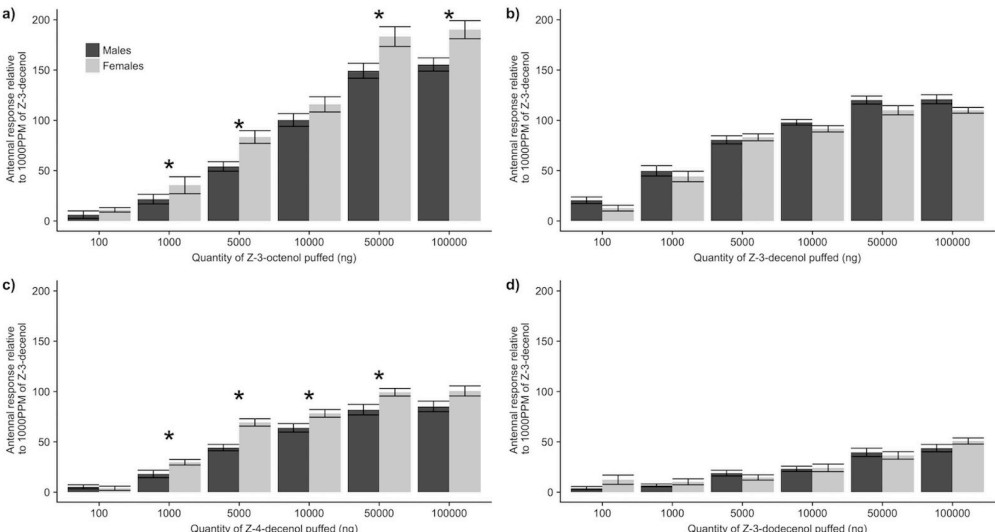

**Fig 2. Electro-antennogram dose-response profiles of four compounds found in male effluvia.** Relative EAG responses (mean ± SE) of a) (*Z*)-3-octenol, b) (*Z*)-3-decenol, c) (*Z*)-4-decenol and d) (*Z*)-3-dodecenol puffed on male (black, n = 10) and female (grey, n = 10) *S. noctilio* antennae for six different doses tested. Asterisk indicates significant differences (p < 0.05) between male and female for the same dose puffed.

antennal sensitivity for doses of 50 000 ng and 100 000 ng were similar for both males (p = 0.597) and females (p = 0.622) and suggest antennal saturation for this compound.

(*Z*)-3-Dodecenol elicited the smallest antennal responses for all doses in both sexes (Fig 2d). Relative antennal responses of males and females were similar for each dose tested.

## Determination of production site of putative male pheromone blend

Three putative pheromone components, (*Z*)-3-decenol previously reported by Cooperband et al. [30], and (*Z*)-3-octenol and (*Z*)-3-dodecenol identified in this study are all released from the hind legs of *S. noctilio* males (Fig 3). Since the three compounds had the same source of emission, all the data were pooled. A significant difference (p < 0.001) was observed in the amount of the 3 putative pheromone components in the head, thorax and abdomen (Fig 3a). A significant difference (p = 0.003) was observed among the different parts of the thorax (Fig 3b). More than 90% of (*Z*)-3-octenol and (*Z*)-3-decenol and more than 35% of (*Z*)-3-dodecenol were released from the hind legs. Small quantities of these putative pheromone components were sometimes detected in samples from other parts of the body and it is suspected that it might be due to male behaviour (e.g., during grooming). Significant differences were not observed (p = 0.114) among the parts of the hind legs (Fig 3c). It was not possible to accurately quantify the amount of (*Z*)-4-decenol as it eluted in the tail of (*Z*)-3-decenol. No trace of (*Z*)-4-decenol was found, likely because it co-elutes with (*Z*)-3-decenol. Investigation of the body parts where the (*Z*)-3-decenol was not found did not reveal traces of (*Z*)-4-decenol, suggesting that the (*Z*)-4-decenol and the (*Z*)-3-decenol are released together.

The SEM study did not reveal any regions with abundant pores on the surface of the hind legs of males. A sexually dimorphic region was found on the dorsal view of the tibia (Fig 4). Both males and females have a triangular shaped structure on the back of their tibia but it is larger on males. The inside of the triangular-shaped structures were smooth on males but porous on females.

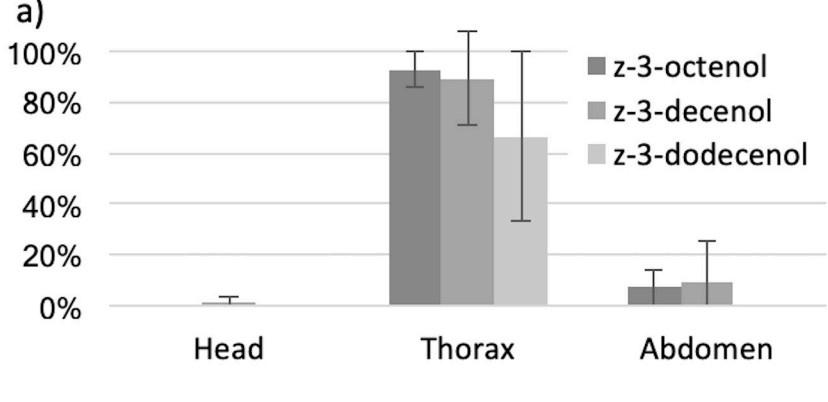

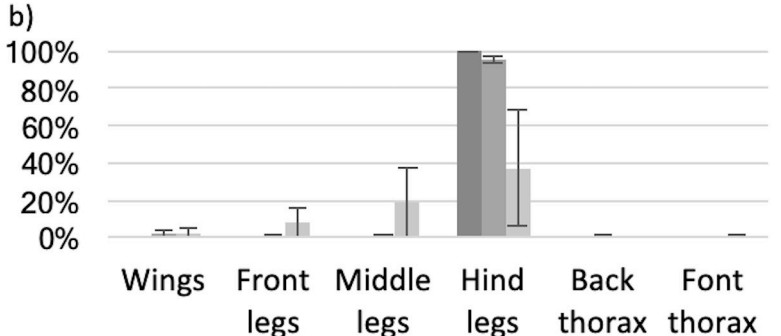

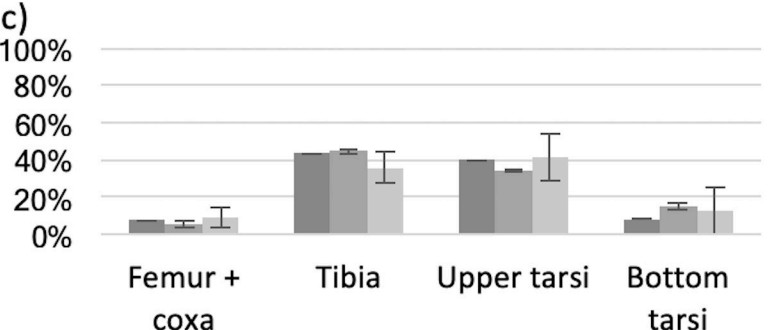

**Fig 3. Confirmation of the origin of the blend of pheromone from the hind legs of males.** Panel a) shows the whole body segment aerations, panel b) aerations of the portions of the thorax and appendages and panel c) the aerations of the portions of the hind legs. Percentage of pheromone released found for each series / mg of dry mass ± SD of (*Z*)-3-octenol (dark grey), (*Z*)-3-decenol (medium grey) and (*Z*)-3-dodecenol (light grey).

## Comparison of putative pheromone release rates: Males and lures

Males of *S. noctilio* were found to release more (*Z*)-3-decenol than any of the three types of lures. The average relative release rate for males was found to be 90 ± 12.4 ng/min of (*Z*)-3-decenol. The average relative release rates after two days were 23.7 (± 7.2), 5.9 (± 2.8) and 19.6 (± 23.7) ng/min respectively for the Silicone, PE and SEPTA lure. Relative release rates decreased to 0.1 (± 0.1), 4.5 (± 4.1) and 3.6 (± 3.6) ng/min after seven days.

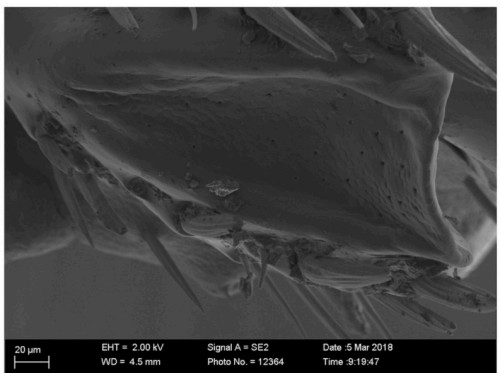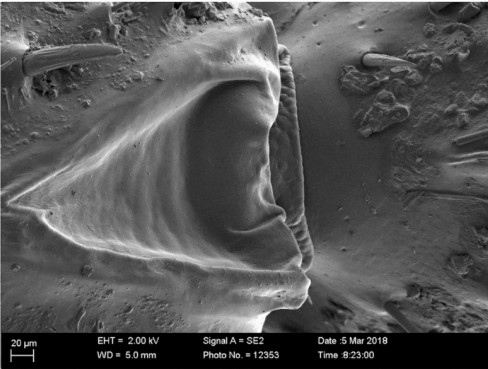

**Fig 4. Scanning electron microscope of the triangle shape situated on the back of the basitarsus of *S. noctilio*.**
Many pores are visible in the triangle shape hind legs of the female (left) that were not visible on males (right).

## Discussion

This study increases our understanding of the chemistry, neurophysiology and site of synthesis or storage of male specific compounds in *S. noctilio*. Two of the four compounds detected in this study, (*Z*)-3-decenol and (*Z*)-4-decenol, were previously described in male *S. noctilio* effluvia [30]. The compounds (*Z*)-3-octenol and (*Z*)-3-dodecenol were identified in male effluvia and this is the first report of these compounds in male *S. noctilio* effluvia. No trace of the other putative pheromone component (*E,E*)-2,4-decadienal [26] was detected from male effluvia in this study. Antennae of both sexes were sensitive to the four compounds, but differences in sensitivity were observed among compounds, sex, and doses tested. Generally, at low and high doses larger antennal responses were elicited by (*Z*)-3-decenol and (*Z*)-3-octenol, respectively, than by any of the other compounds. At most doses tested, there were no differences in the size of male and female responses; however, for (Z)-3-octenol and (*Z*)-4-decenol some differences were observed. In all of these instances female responses were larger than male responses. The site of synthesis and/or storage was determined to be the hind legs in males. The relative release rate of (*Z*)-3-decenol from the hind legs was determined to be 90 ± 12.4 ng/min per male, 4–15 times higher than any of the three lures tested. The actual release rate from males is likely higher than measured here because only a fraction of the released compound absorbs into the SPME fibre.

Both the chemistry and emitting sex of the male specific compounds in *S. noctilio* differ from the general model known for non-apocrita Hymenoptera. All known sex pheromones in non-apocrita Hymenoptera have been reported to be female emitted. Sex pheromones in Diprionidae species possess acetate and/or propanoate of di- and tri- methyl-branched 2-alkanol of 11–20 carbon long backbones with different chiral centres [40–50]. In Cephidae, 9-acetyloxynonanal in *Cephus cinctus* [51] and (9*Z*)-octadec-9-en-4-olide in *Janus integer* [52] were described as sex pheromones. The sex pheromone in *Acantholyda erythrocephala* (Pamphiliisdae) was identified as (*Z*)-6,14-pentadecadienal [53] and the sex pheromone in *Pikonema alaskensis* (Nematidae) is a blend of (*Z*)-5-tetradecen-l-ol and (*Z*)-10-nonadecenal [42, 54]. Sex pheromones in sawflies are mostly produced from the oxidation of cuticular hydrocarbons [e.g., *A. erythrocephala* [53], *P. alaskensis* [54], *Macrocentrus grandii* [55], *C. cinctus* [51]] and these type of compounds were not observed in *S. noctilio*.

The chemistry of the male specific compounds in *S. noctilio* is similar to that found in various insects and fungi. (*Z*)-3-Octenol has previously been described as a larval pheromone in the sawfly *Hoplocampa testudinea* (Hymenoptera: Tenthredinidae) [56]. 3-Octenol was also

reported to be produced by a species of fungus in the genus *Mucor* [57]. Different *n*-octenol molecules were reported to be emitted from Basidiomycete mushrooms and in mushroom-mimicking orchids that mimic volatiles associated with the oviposition site of fungus gnats (Diptera) [58, 59]. (*Z*)-3-Decenol is a minor component of the aggregation pheromone of the Cerambicid beetle *Rosalia funebris* [60]. (*Z*)-3-Dodecenol is a trail pheromone in the Kalotermitidae and Macrotermitinae families of termites [61–63]. Independently, 3-octenol, (*Z*)-3-decenol and (*Z*)-3-dodecenol are attractive to insects from different orders.

The emission source of the male specific compounds was found to be the sexually dimorphic hind legs in *S. noctilio*. No significant differences of the normalised quantity of the male specific compounds present between the different parts of the hind legs were recorded. A possible explanation is that the male specific compounds are stored in the leg tendon gland of the hind legs. The leg tendon gland is a hollow reservoir starting in the femur and running down to the unguitractor plate, manubrium and arolium structures situated at the end of the clamp [64–66]. The arolium structures are likely to be used in pheromone release and are missing or very reduced in *S. noctilio* females but well developed in males [67]. In our experimental design, we cut the hind legs into different parts, which probably allowed the male specific compounds to be released from the different hind leg parts and not only from the terminal opening. Use of the leg tendon gland for storage would allow males to contain large amounts of compounds and could explain the sexual dimorphism of the hind legs and terminal clamp structures in *S. noctilio*.

The hind leg gland plays an important role in communication in the Hymenoptera. Hind legs in ants are known or hypothesized to release trail, sex, lek-formation and marking pheromones. For example, the genus of ant *Crematogaster* is known to release trail pheromone from the tibial tendon gland of their hind legs [68]. The major component of the trail pheromone of *C. castanea* was identified as (*R*)-2-dodecanol [69]. Females in *Ascogaster reticulatus* release (*Z*)-9-hexadecenal as a sex pheromone from the hind legs [70] and male *Polistes dominulus* are thought to release a lek-formation pheromone from their legs [71]. Foragers in the stingless bee *Melipona seminigra* use a pheromone from its hind leg tendon gland to mark its food site [66]. The similarity in chemistry of compounds released from the hind legs of *S. noctilio* and other Hymenoptera suggest that these compounds may mediate similar behaviours in *S. noctilio*.

Two enlarged glomeruli were found in the male antennal lobe of *S. noctilio* (J. Spaethe, personal communication). Different antennal response amplitudes were shown to be linked to glomeruli size, and are typically linked to specific olfactory receptors [72]. Sexually dimorphic enlarged glomeruli are typically linked to pheromone processing [73–75]. For example, the processing of the queen pheromone in the honeybee was linked to enlarged glomeruli in males [73]. This pheromone elicits an antennal response in males that is 2.5 times bigger than in workers [76]. Our dose-response experiment did not show a higher antennal sensitivity in males compared to females for the four compounds tested. The antennal recovery time in our study was a compromise between recovery between puffs and the length of the experiment. A longer (and likely more complete) recovery would have meant longer experiments and increased risk of loss of sensitivity due to degradation of the antennal preparation. It is possible that our recovery time led to an underestimation of the antennal response, but our randomisation of treatments would have prevented any bias. This result suggests that the enlarged glomeruli found in males are not involved in the detection of the male produced compounds that we tested here, and must be linked to other compounds that are differentially detected between males and females.

The additional information provided by this study on the male specific compound blend released by *S. noctilio* corresponds to the description of an aggregation-sex pheromone [77].

Aggregation-sex pheromones typically elicit a similar response from both male and female antennae [78–81]. Additionally, aggregation behaviours are usually induced by chemicals released by males [82–84]. In many insects lek and trail pheromones are released from the hind legs. Cumulatively these observations are consistent with the hypothesis that the male produced compounds reported here mediate lek formation in males and lek location in female *S. noctilio*.

Various factors such as the pheromone blend composition or lure release rate can influence pheromone activity in field trials. Female *J. integer* were shown to release ~10 ng/female/day [52]. Traps captured more *J. integer* when baited with higher quantity of pheromone [85]. None of the lures tested in this study could match the average relative release rate of male *S. noctilio*. If the lures previously tested in the field [25, 31] and those sampled in this study have similar release rates, their low release rate might have contributed to the negative results. It is also possible, as suggested by the differences in chemistry and the releasing sex, that these compounds may not function as a sex pheromone in *S. noctilio*. In addition to aggregation, male-produced pheromones in Hymenoptera also mediate dispersal, territory marking and aggression [45, 86].

The production and storage of large quantities and ability of both sexes to detect the compounds identified in male hind legs, suggests that one or more of these compounds may play a role in woodwasp biology. Comparison of our results with the existing literature suggests that if these compounds are sex pheromones in *S. noctilio* they would differ from all other known sawfly pheromones in terms of chemistry and the releasing sex. Our results are consistent with the hypothesis that these male released compounds mediate lek formation and location in males and females, respectively.

## Supporting information

**S1 Fig. Three dimensional chromatogram showing the two co-eluting peaks (A and B) eluting in the tail of the large major chromatographic peak released by males.** The best tentative match for peak A was (*Z*)-3-decenol and peak B was (*Z*)-4-decenol in the NIST library.
(TIFF)

**S2 Fig. Antennal recovery (in %) 30, 45 and 60 sec after 10 000 ng of (*Z*)-3-decenol puffed (n = 3 males and 3 females).** Different letters indicate significant differences of the recovery time.
(TIFF)

## Acknowledgments

We thank the South African forestry company SAPPI and Mr Phillip Croft from the Institute for Commercial Forestry Research (ICFR), Denzil Lawrie and his team for providing infested logs. We also thank the department of Chemistry at the University of Pretoria, especially Dr. Yvette Naudé for the use of the GC X GC TOF MS and Dr. Divan van Greunen for synthesising the major pheromone compound. Finally, we thank Mrs. Erna van Wilpe and her team for the use of the scanning electron microscope at the University of Pretoria.

## Author Contributions

**Conceptualization:** Quentin Guignard, Marc Bouwer, Bernard Slippers, Jeremy Allison.

**Data curation:** Quentin Guignard.

**Formal analysis:** Quentin Guignard.

**Funding acquisition:** Bernard Slippers, Jeremy Allison.

**Investigation:** Quentin Guignard.

**Methodology:** Quentin Guignard, Marc Bouwer, Jeremy Allison.

**Project administration:** Bernard Slippers, Jeremy Allison.

**Resources:** Bernard Slippers, Jeremy Allison.

**Supervision:** Bernard Slippers, Jeremy Allison.

**Visualization:** Quentin Guignard.

**Writing – original draft:** Quentin Guignard.

**Writing – review & editing:** Marc Bouwer, Bernard Slippers, Jeremy Allison.

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
