## [Decision Letter · Decision Letter 0]

23 Oct 2020

PONE-D-20-30017

The putative male aggregation-sex pheromone in *Sirex noctilio* (Hymenoptera: Siricidae)

PLOS ONE

Dear Dr. Allison,

Thank you for submitting your manuscript to PLOS ONE. After careful consideration, we feel that it has merit but does not fully meet PLOS ONE’s publication criteria as it currently stands. Therefore, we invite you to submit a revised version of the manuscript that addresses the points raised during the review process.

All three reviewers felt the manuscript has strong value and should be publishable.  Reviewers #1 and #2 went very carefully through your manuscript finding places where the manuscript can be improved for clarity, reproducibility, and in some cases to address the validity of conclusions.  The reviewers listed several areas where additional experiments would be helpful.  However, overall, I believe the manuscript can be made acceptable through attention to their comments and adding exposition in the methods, and clarifying the study limitations in the discussion.  Comments that need close and complete attending to include all six major points from reviewer #1, with especially close attention paid to Reviewers comment #1, where the reviewer discusses doses of the pheromone blends and how they were delivered - in comparison to commercial lures.  Many of the other major comments are also addressed by Reviewer #2 desire for more detail in the methods or more thorough discussion of the limits of this study.    

Reviewer #2  requested a title change. I think this could increase your readership and is a good suggestion, but I do not think it is necessary.  

The reviewers also offered many suggested Line edits that I find to be very useful.  

We look forward to receiving your revised manuscript.

Kind regards,

Gregg Roman, PhD

Academic Editor

PLOS ONE

Journal Requirements:

2.We note that you have indicated that data from this study are available upon request. PLOS only allows data to be available upon request if there are legal or ethical restrictions on sharing data publicly. For more information on unacceptable data access restrictions, please see http://journals.plos.org/plosone/s/data-availability#loc-unacceptable-data-access-restrictions.

Reviewers' comments:

Reviewer's Responses to Questions

**Comments to the Author**

1. Is the manuscript technically sound, and do the data support the conclusions?

Reviewer #1: Yes

Reviewer #2: Yes

Reviewer #3: Yes

2. Has the statistical analysis been performed appropriately and rigorously? 

Reviewer #1: Yes

Reviewer #2: Yes

Reviewer #3: I Don't Know

3. Have the authors made all data underlying the findings in their manuscript fully available?

Reviewer #1: No

Reviewer #2: Yes

Reviewer #3: Yes

4. Is the manuscript presented in an intelligible fashion and written in standard English?

Reviewer #1: Yes

Reviewer #2: Yes

Reviewer #3: Yes

5. Review Comments to the Author

Reviewer #1: A male-produced pheromone has been identified in Sirex noctilio, but its bioactivity has been equivocal and field-activity has yet to be demonstrated. Unfortunately, this paper does not address this important issue. Instead, the authors confirm the presence of some male-specific components, fail to find others, and report new components. They show EAG bioactivity, the site of accumulation of the pheromone (male hind legs), and presence of these compounds in headspace collections. Thus, a new blend of four compounds includes (Z)-3-decenol, (Z)-4-decenol, (Z)-3-octenol and (Z)-3-dodecenol, plus compounds previously identified that they were unable to recover in this study.

This is a good paper, hopefully representing another incremental step toward identifying a blend that could be used to monitor Sirex populations. I think the paper merits publication with some rather minor edits to clarify some points.

1. The work comparing Sirex release rates to release rates from lures is particularly inconclusive. The authors compare home-made lures to release rates from live Sirex. First, are there commercial lures, and if so why are they not used. Second, no one loads technical material of any sort without solvent into rubber septa. The authors load 10 ul (10 mg) directly into septa. This is incredibly high, and without solvent, all of the reagent stays at the surface of the rubber and rapidly evaporates. Third, the septa are allowed to sit in the fume hood for 2 days, during which time most of that 10 mg is rapidly evaporated. It is not surprising that the release rate at 2 and 7 days is low. Fourth, generally formulations are engineered to mimic the release rate of the insect. It is not clear on what basis the current home-made formulations were made. Finally, it is not clear why silicone is referred to a Teflon tube and whether the 0.4 ml microcentrifuge tube is capped. In L569 it is not clear to me what the basis is for saying that previously used lures “likely had release rates similar to those of lures observed in this study”. Overall, this is a weak section of the paper.

2. L198-199, Fig. S2: Very large doses are being used to determine recovery of the antenna in 15 sec increments. The results (Fig. S2) show a recovery trend that has not yet reached an asymptote. I think at such high doses, a longer recover (2 or more min) is likely needed. Might mention this as a limitation in the Discussion.

3. L212-217: The EAG description is lacking details, compared to other parts of this manuscript: what was the flow rate, puff duration, amplifier filtering (high and low pass filters)?

4. Table 1: All the entries are also elaborated in the text. Is the table needed? Or alternatively reduce the redundancy in text. Also change “,” to “.”

5. L5104: Extremely slippery ground to go from EAG responses to "totally or partially different receptors". The EAG tells you nothing about what receptors are involved unless you conduct cross-adaptation experiments. This section is unnecessarily way too speculative. Receptors are not part of this investigation.

6. L528-534: This paragraph summarizes where the pheromone compounds are found outside Sirex. Why go to fungi and mushrooms, when 3-8:OH is a pheromone in sawfly larvae (Boevé, J.-L., Dettner, K., Francke, W., Meyer, H., and Pasteels, J.M. 1992. The secretion of the ventral glands in Nematus sawfly larvae. Biochem. Syst. Ecol. 20:107-111), and 8:OH is found in dozens of insect species (see Pherobase).

L1 title: Replace “The” with “A”

L31: “A blend of four compounds was identified”

L74: “Subsequent field trials using this pheromone blend with and without pine volatiles observed no effect on the number of females captured”. Be more direct and explicit. Did it trap females, but at numbers no different from unbaited traps? Did it fail to attract females? This is pretty critical to the significance of this work.

L89: Refrigerator

L104 and elsewhere: Septum, not septa for singular

L136: CaCl2

L167: You are derivatizing primary alcohols, not hydrocarbons, unless you mean the aliphatic chain of the alcohol.

L172 and elsewhere: The use of ppm is unconventional in chemical ecology. It’s fine as is, but I’d recommend converting to ng/ul.

L294: The statement “The chromatograms of samples obtained from female wasps were identical to blank samples.” Concerns me. I have never seen a headspace collection over any insect be identical to blanks. Maybe the authors mean to say that there were no unique compounds that were female-specific relative to males?

L296 and elsewhere: Index values or indices.

L512: Ref #41 is on moths. The authors cite honeybees presumably because, like Sirex, they are hymenopterans. But the connection between size of glomeruli and sexually dimorphic pheromone processing is best understood in moths, not bees.

L541: release

L552: Polistes

Reviewer #2: This paper looks at the production site of the putative male-produced pheromone of Sirex noctilio as well as release rates compared to lures used in trapping. The authors also looked at the antennal response to two new chemicals found in this pheromone. In general, the paper is well-written and the methods match up to the intended goals and objectives. I do have several concerns that I hope will help the authors in revisions:

First, the title of the paper is not very descriptive. It should reflect that the authors were looking more at the biology/ecology of the pheromone.

There are several places in the introduction that could be expanded and have citations added. L47 and L50 both need citations to support the claims. In L48, the authors state that larvae bore into the sapwood - larvae can bore all the way into the heartwood of trees (again needs a citation). The end of L65 also needs a citation.

The introduction could be better integrated. For example, the paragraphs "control measures..." and "olfactory attractants..." seem disjunct. I think moving L61-62 up to the beginning of its respective paragraph and mentioning other parasites (like Ibalia) and their attraction to fungal cues could make this integration better. There's also no mention of parthenogenesis which seems important considering the lack of female response discussed later (this applies to the discussion as well).

I also have several issues with the methods as they're presented. A lot of basic information is either missing or shows up much too late. When were logs collected? How long were they? Over what time period were they collected? How long were wasps stored in the fridge? Did you have an end time at which they were not usable? Did this storage prevent any of them from being usable?

I think that it would serve the methods to state right off the bat the number of wasps and other basic info. This information is presented in the first paragraph of the results but I think it would be better to have it in the methods. I found myself questioning how many or how long for several of your methods. Without this information the methods were confusing to read and I don't think reproducible.

Minor methods comment:

L144: what was the set of algorithms? How were they determined?

How were the puff amounts determined? Is this standard? If so, state that.

What determined whether a test (all of them) was successful? Did you have to remove any data or samples? Were they all successful?

L285: were these data not normal? The only normality/variance testing you mentioned was normal with equal variance so I'm not sure why you're using a KW test.

replace Kovat retention index with KI after the first mention (you bounce back and forth several times).

replace ml with mL throughout the paper.

L312: the end parentheses is missing.

L405: remove extra spaces in 100:1

L435: what's the actual p value?

Some parts of the discussion seemed over the top and unnecessary (i.e. L517-526). Additionally, this section is a bit disjunct like the intro. L507-515 seem unnecessary the way its laid out and the tie in is much later and after other paragraphs of info.

Minor discussion comments:

L508: remove extra space before 'glomeruli'

L529: 'sp' is not necessary (and isn't italicized) when you're discussing a genus with multiple species as the multiple species are implied.

L563: add 's' after 'male'

L578: add 'that' between 'hypothesis' and 'these'

Reviewer #3: This manuscript describes experiments conducted to determine the identity of pheromones emitted by the woodwasp, Sirex, noctilio. The experiments conducted and analysis of volatiles and extracts collected has positively confirmed that two compounds are produced by males and that both males and females can detect these compounds. They also determined that male hind legs are the probable source of these compounds. The relative release rate from the male wasps was also determined. This release rate from males was shown to be greater than that from lures used previously. This could be the reason that lures are not attracting great numbers of woodwasps. It would have been nice to test the hypothesis that a greater release rate from lures would attract more woodwasps of both sexes. The lures may not actually trap the wasps but they would be found in increased numbers around the lure. This is a well written manuscript and could be published without modification. Figure 3 has a typo – tarse = tarsi

6. PLOS authors have the option to publish the peer review history of their article (what does this mean?). If published, this will include your full peer review and any attached files.

Reviewer #1: No

Reviewer #2: No

Reviewer #3: No

---

## [Author Response · Author response to Decision Letter 0]

19 Nov 2020

As outlined in the response to reviewers letter we have responded to all six major points from reviewer #1, with especially close attention paid to this Reviewers comment #1, where the reviewer discusses doses of the pheromone blends and how they were delivered - in comparison to commercial lures. We also explicitly addressed the desire of Reviewer #2 for more detail in the methods or more thorough discussion of the limits of this study.

---

## [Decision Letter · Decision Letter 1]

8 Dec 2020

PONE-D-20-30017R1

Biology of a putative male aggregation-sex pheromone in Sirex noctilio (Hymenoptera: Siricidae)

PLOS ONE

Dear Dr. Allison,

Thank you for submitting your manuscript to PLOS ONE. After careful consideration, we feel that it has merit but does not fully meet PLOS ONE’s publication criteria as it currently stands. Therefore, we invite you to submit a revised version of the manuscript that addresses the points raised during the review process.

Both reviewers were appreciative of the improvements made to this manuscript in revision.  Both reviewers closely read the manuscript again and identified further minor issues with language that I would like you to address before publication.   The reviewers have outlined this comments clearly.  Please pay close attention to these comments and make the necessary changes.

I also want to highlight the question reviewer number 1 has regarding the use of solvent in loading the lures.  As this reviewer points out, it is hard to know exactly what you have done in this procedure.  Without clearly stating what solvents were used and how much, these experiments are not repeatable.   Moreover, one's interpretation of the results strongly depend on how the septa were loaded and the solvent conditions. This section, around line 270, needs to be much more clear prior to publication.    

We look forward to receiving your revised manuscript.

Kind regards,

Gregg Roman, PhD

Academic Editor

PLOS ONE

Reviewers' comments:

Reviewer's Responses to Questions

**Comments to the Author**

1. If the authors have adequately addressed your comments raised in a previous round of review and you feel that this manuscript is now acceptable for publication, you may indicate that here to bypass the “Comments to the Author” section, enter your conflict of interest statement in the “Confidential to Editor” section, and submit your "Accept" recommendation.

Reviewer #1: (No Response)

Reviewer #2: All comments have been addressed

2. Is the manuscript technically sound, and do the data support the conclusions?

Reviewer #1: Yes

Reviewer #2: Yes

3. Has the statistical analysis been performed appropriately and rigorously? 

Reviewer #1: Yes

Reviewer #2: Yes

4. Have the authors made all data underlying the findings in their manuscript fully available?

Reviewer #1: Yes

Reviewer #2: Yes

5. Is the manuscript presented in an intelligible fashion and written in standard English?

Reviewer #1: Yes

Reviewer #2: Yes

6. Review Comments to the Author

Reviewer #1: In general, the paper has improved, as the authors carefully revised the manuscript in response to comments. However, I am still perplexed by the unwillingness to directly respond to simple questions. Some of this may be related to the authors’ unusual responses of paraphrasing reviewer comments rather than responding directly to them.

For example, I stated in the earlier review: “no one loads technical material of any sort without solvent into rubber septa. The authors load 10 ul (10 mg) directly into septa. This is incredibly high, and without solvent, all of the reagent stays at the surface of the rubber and rapidly evaporates. … the septa are allowed to sit in the fume hood for 2 days, during which time most of that 10 mg is rapidly evaporated. It is not surprising that the release rate at 2 and 7 days is low.”

The authors respond with “The reviewer asked if we loaded solvent with the pheromone in the red rubber lure. Yes. We clarified how we loaded the red rubber septa with solvent in lines 273-275.”

But the revised manuscript states (L269): “Three lures of each type were filled with 10 μL of (Z)-3-decenol (95% purity)”. Since there is no mention of solvent in subsequent lines, to me this means that 10 uL (~10 mg) of pure alcohol (neat, technical, no solvent) was loaded without solvent directly into the septum.

As written, no other researcher would be able to repeat this experiment unless the authors clarify exactly what they did: how much (Z)-3-decenol (mg) was loaded into what volume (uL) of which solvent (hexane?) and then loaded into the septum? Therefore, this is not a trivial question and it should have been properly addressed in the revision.

L143: CaCl2 is still not properly presented. 2 should be subscripted.

L272: “Themo Fisher” misspelled.

Reviewer #2: Overall, the authors did a good job of addressing reviewers’ comments. After re-reading, I agree with the author’s decisions to keep L496-505. I like the additions to the intro and discussion and think they strengthened the paper significantly. I do have several minor comments detailed below.

There are still a couple places where Kovat’s retention index is written out and it should be KI throughout the paper once defined (e.g. L180, L307).

There are many missing commas that make readability difficult sometimes:

L43: comma before/after Sirex noctilio

L46: I think this sentence would make more sense as “…inject its eggs, a phytotoxic venom, and the symbiotic fungus…” As it reads now it makes it seem like the fungus has eggs.

L47: commas around “and growth of”

L59: commas around “…and monitor population dynamics of,…”

L60-61: commas around “…and be attracted to…” (I would also delete ‘to’ in front of ‘be attracted to’)

L236: comma between ‘thorax’ and ‘and’

Other minor comments:

L68: I feel like them being parthenogenetic should at least be mentioned. I see the connection between delayed mating and lower fitness, but they are able to produce young without mating. I think without at least mentioning it, this connection isn’t as strong.

L79: the study did not do the experiments, you did 

L253: italicize Sirex

L287: Should be “Student’s” t test

L296: what’s the citation for the chemcal package? And RStudio is just the interface, R is the actual computing language and should be cited.

L299: Is this a new paragraph? Seems like it should just be a continuation from the last one.

L307-308: Rather than state that the results are in table 1, summarize the results and add (table 1). I think you could even just add (Table 1) to the line before and be fine to delete this sentence.

L333-334: now that you’ve clarified in the methods, I think you can delete this sentence as it’s methods material.

L340-344: you should define the () when you talk about the upcoming m/z results.

L370-371: “The large peak was present in very high quantities”

L372: peak should be plural.

L381: was feasible.

L404: was ~100-1000 times…

L477-478: Having SE in () would make this easier to read I think.

L487: comma after ‘sex’

7. PLOS authors have the option to publish the peer review history of their article (what does this mean?). If published, this will include your full peer review and any attached files.

Reviewer #1: No

Reviewer #2: No

---

## [Author Response · Author response to Decision Letter 1]

14 Dec 2020

We have tried to clarify our description of how the lures, specifically the septa, we loaded. We hope that it is clear now. In short, I agree with the reviewer that standard practice is to load the pheromone in solvent. That was not done. Rather, immediately after removing the septa from solvent, the septa were loaded with neat pheromone and left in a fume hood to allow the solvent to evaporate. As the solvent evaporated the pheromone would have further penetrated the septa. We think how the lures were loaded is clear now. Thank you.

---

## [Editor Report · Decision Letter 2]

21 Dec 2020

Biology of a putative male aggregation-sex pheromone in Sirex noctilio (Hymenoptera: Siricidae)

PONE-D-20-30017R2

Dear Dr. Allison,

We’re pleased to inform you that your manuscript has been judged scientifically suitable for publication and will be formally accepted for publication once it meets all outstanding technical requirements.

Kind regards,

Gregg Roman, PhD

Academic Editor

PLOS ONE
---

## [Editor Report · Acceptance letter]

23 Dec 2020

PONE-D-20-30017R2 

Biology of a putative male aggregation-sex pheromone in Sirex noctilio (Hymenoptera: Siricidae) 

Dear Dr. Allison:

I'm pleased to inform you that your manuscript has been deemed suitable for publication in PLOS ONE. Congratulations! Your manuscript is now with our production department. 

Kind regards, 

on behalf of

Dr Gregg Roman 

Academic Editor

PLOS ONE